# Antibacterial Activity of Nanostructured Zinc Oxide Tetrapods

**DOI:** 10.3390/ijms24043444

**Published:** 2023-02-08

**Authors:** Aike Büter, Gregor Maschkowitz, Martina Baum, Yogendra Kumar Mishra, Leonard Siebert, Rainer Adelung, Helmut Fickenscher

**Affiliations:** 1Institute for Infection Medicine, University Medical Center Schleswig-Holstein, Christian-Albrecht University of Kiel, 24105 Kiel, Germany; 2Functional Nanomaterials, Institute of Material Science, Christian-Albrecht University of Kiel, 24143 Kiel, Germany; 3Smart Materials, NanoSYD, Mads Clausen Institute, University of Southern Denmark, 6400 Sønderburg, Denmark; 4Kiel Nano, Surface and Interface Science (KiNSIS), Christian-Albrecht University of Kiel, 24118 Kiel, Germany

**Keywords:** zinc oxide tetrapods, *Staphylococcus aureus*, *Klebsiella pneumoniae*

## Abstract

Zinc oxide (ZnO) tetrapods as microparticles with nanostructured surfaces show peculiar physical properties and anti-infective activities. The aim of this study was to investigate the antibacterial and bactericidal properties of ZnO tetrapods in comparison to spherical, unstructured ZnO particles. Additionally, killing rates of either methylene blue-treated or untreated tetrapods and spherical ZnO particles for Gram-negative and Gram-positive bacteria species were determined. ZnO tetrapods showed considerable bactericidal activity against *Staphylococcus aureus,* and *Klebsiella pneumoniae* isolates, including multi-resistant strains, while *Pseudomonas aeruginosa* and *Enterococcus faecalis* remained unaffected. Almost complete elimination was reached after 24 h for *Staphylococcus aureus* at 0.5 mg/mL and *Klebsiella pneumoniae* at 0.25 mg/mL. Surface modifications of spherical ZnO particles by treatment with methylene blue even improved the antibacterial activity against *Staphylococcus aureus*. Nanostructured surfaces of ZnO particles provide active and modifiable interfaces for the contact with and killing of bacteria. The application of solid state chemistry, i.e., the direct matter-to-matter interaction between active agent and bacterium, in the form of ZnO tetrapods and non-soluble ZnO particles, can add an additional principle to the spectrum of antibacterial mechanisms, which is, in contrast to soluble antibiotics, depending on the direct local contact with the microorganisms on tissue or material surfaces.

## 1. Introduction

Nanomaterials, in general, are under investigation for the control and therapy of infections, such as wound infections. For example, specific crystalline molybdenum disulfide nanomaterials display high photothermal performance and exhibit strong antibacterial effects [1,2]. In addition, facet-dependent gold nanocrystals were capable of an effective photothermal killing of bacteria [3].

Zinc oxide tetrapods (t-ZnO) are micrometer-sized particles consisting of four monocrystalline ZnO arms with nano-structured surfaces. They are widely investigated in research fields such as electronics [4], sensing devices [5], composites or surface coatings [6,7], energy harvesting [8], and biomedical applications [9]. The physical properties of ZnO particles were studied in depth in previous publications, such as by Raman spectroscopy, X-ray diffraction, and photoluminescence. They exhibit a very high degree of crystallinity, proven in previous publications by both Raman and X-ray diffraction and, therefore, the resulting semiconducting properties are excellent, as can be determined with photoluminescence [10,11,12]. X-ray diffraction, for instance, showed especially the characteristic (100), (002), and (101) reflections, which one would expect for zinc oxide that crystallizes in the wurtzite-type structure [12]. The investigation also showed no significant variations in the crystal quality of the zinc oxide tetrapods, which was confirmed by the Raman spectra. They repeatedly revealed the standard peaks at 100 cm^−1^, 333 cm^−1^, and 438 cm^−1^ that are related to oxygen and zinc sublattices in the crystal structure of ZnO [13]. A review article on the properties and the broad application research range is available [14].

The size of the tetrapods (length of the arms approximately 30 µm, diameter approximately 1 µm) makes them attractive for direct cell contact, as the arms cannot penetrate into eukaryotic cells and destroy them from inside. Instead, surface adsorption and immune triggering mechanism was observed in the case of the non-bacterial herpes simplex virus [15]. However, the surface-dependent antibacterial mechanisms of t-ZnO remain to be further elucidated. A release of reactive oxygen species or Zn^2+^ ions from the surface of the t-ZnO has been proposed [9], and t-ZnO is much less toxic to normal human fibroblasts when compared to other Zn sources [16].

Spherical ZnO particles (s-ZnO) are much smaller than t-ZnO, can more easily release Zn^2+^ ions, and cause apoptosis in various cell types [17,18,19]. They typically occur as particles in a size of less than 1 µm (Figure 1B). Due to their higher release of Zn^2+^ ions and toxicity, they provide a narrow therapeutic window only. In comparison, t-ZnO have been successfully used as a cell-sparing antibacterial agent embedded into polymeric materials for various medical and even antibacterial applications [9,20,21]. t-ZnO are microparticles with four protruding, monocrystalline arms showing specific physical properties of their nanosurfaces, which can be chemically modified. Such a process can expose other crystal facets with varying antibacterial activity. Compared to the much smaller and less crystalline s-ZnO particles, t-ZnO tetrapods show massively weaker toxicity in eukaryotic cell culture. Since t-ZnO and s-ZnO particles are not soluble, they need to be applied in suspension and may be useful for local treatment, such as in wound infections. The aim of this study was to investigate the antibacterial and bactericidal properties of t-ZnO in comparison to commercially available s-ZnO. Killing rates of microscale tetrapods and smaller s-ZnO particles against selected Gram-negative and Gram-positive bacteria species were determined for different time points and concentrations.

## 2. Results

The t-ZnO and s-ZnO particles used in this study were further analyzed for their structure by electron microscopy (Figure 1). The tetrapod structure of t-ZnO was clearly observed, in contrast to the less structured, spherical form of s-ZnO. Thus, the documented morphology confirmed the expected structures (e.g., [9,22,23]).

The antibacterial activity of t-ZnO particles was determined both against selected clinically relevant Gram-negative and Gram-positive bacteria of the species *Klebsiella* (*K.*) *pneumoniae, Staphylococcus* (*S.*) *aureus, Pseudomonas* (*P.*) *aeruginosa*, and *Enterococcus* (*E.*) *faecalis* in orientating tests in triplicate independent values (Figure 2). The antibacterial activity was measured after 1, 3, or 24 h of incubation, covering the major time points of microbiological relevance. Since antibacterial effects were detectable for *Klebsiella* (*K.*) *pneumoniae* and *Staphylococcus* (*S.*) *aureus,* these species were further analyzed and, in addition to reference strains, also with laboratory isolates of multi-resistant bacteria, such as methicillin-resistant *S. aureus* or carbapenem-resistant *K. pneumoniae* strains.

### 2.1. Activity against Staphylococcus aureus

Screening for bactericidal activity of t-ZnO against methicillin-sensitive *S. aureus* showed a clear dose- and time-dependent bactericidal effect (Figure 2B).

With increasing concentrations, s-ZnO particles were able to achieve a nearly complete killing after 24 h with 2 mg/mL (Figure 3A). The ZnO tetrapods were able to reduce bacterial numbers by three orders of magnitude at all concentrations after 24 h. At the lowest concentration (0.5 mg/mL) of t-ZnO, bacterial counts were reduced by two orders of magnitude after 24 h of incubation. The experiments with 1 mg/mL ZnO tetrapods demonstrated comparable effects with 0.5 and 2 mg/mL and were, thus, largely independent of the dosage. s-ZnO caused a reduction of approximately four orders of magnitude after 24 h. After 24 h incubation, methylene blue-treated and untreated s-ZnO particles reduced bacterial counts by approximately three orders of magnitude at 0.5 mg/mL and 1 and 2 mg/mL eradicated the bacteria almost fully as compared to the s-ZnO particles. Both the treated and untreated tetrapods exhibited more surviving bacteria after 24 h in comparison to s-ZnO. Methylene blue-treated and untreated t-ZnO exhibited similar antibacterial effects. Incubation with 0.5, 1, or 2 mg/mL methylene blue-treated or untreated t-ZnO resulted in a reduction of approximately two orders of magnitude. Tetrapods were less active than s-ZnO, with increasing activity over time (Figure 3).

Bactericidal activity with different concentrations of untreated or methylene blue-treated s-ZnO particles or tetrapods was also determined against methicillin-resistant *S. aureus* (MRSA). Treatment of MRSA for 24 h with methylene blue-treated and untreated t-ZnO demonstrated a similar antibacterial effect compared to methicillin-susceptible *S. aureus* (Figure 3B). In contrast to the untreated s-ZnO particles, the methylene blue-treated s-ZnO particles showed even a complete eradication of MRSA. The untreated s-ZnO particles reduced bacterial counts by one order of magnitude for 0.5 mg/mL and two orders of magnitude for 1 mg/mL, respectively. While the s-ZnO showed a concentration- and time-dependent antibacterial effect, the t-ZnO effects did not seem to depend on the concentration. All tested concentrations of t-ZnO had a similar effect on the reduction of bacterial counts after 24 h. It appears that the efficacy of t-ZnO does not change within the first 24 h.

### 2.2. Activity against Klebsiella pneumoniae

After 3 h of incubation with untreated s-ZnO particles, there was only a low reduction in bacterial numbers (Figure 4A), while the methylene blue-treated s-ZnO particles slightly increased the reduction levels in bacterial colony numbers. Untreated and methylene blue-treated t-ZnO also caused a reduction of an order of magnitude in the bacterial count after 3 h. Untreated s-ZnO particles displayed a reduction of four orders of magnitude or full eradication after 24 h, whereas the methylene blue-treated s-ZnO particles achieved the complete eradication of bacteria only with the highest concentration. After 24 h, experiments showed complete eradication of *K. pneumoniae* even at the lowest t-ZnO concentration (0.25 mg/mL). Corresponding results were obtained for 0.5 and 1 mg/mL after 24 h of incubation. After 24 h incubation with 0.25 mg/mL untreated and methylene blue-treated t-ZnO, the bacteria counts showed a reduction by almost five orders of magnitude. After 24 h of incubation with 1 mg/mL, untreated or methylene blue-treated t-ZnO demonstrated complete killing of bacteria. Methylene blue-treated t-ZnO displayed a reduction of approximately five orders of magnitude after this time (Figure 4).

Treatment of the KPC3 carbapenemase-producing *K. pneumoniae* strain NTC 13438 with all four different ZnO preparations (Figure 4B) showed only a weak reduction in the number of bacteria after 3 h. After 24 h, all three concentrations of untreated s-ZnO particles exhibited complete eradication of *K. pneumoniae* NTC 13438. Methylene blue-treated s-ZnO particles caused a weak reduction after 24 h at 0.25 mg/mL and a two order of magnitude reduction at 0.5 mg/mL and complete eradication at 1 mg/mL. Treatment of multidrug-resistant *K. pneumoniae* with t-ZnO (Figure 4B) showed only a weak reduction in bacterial numbers at all concentrations compared to treatment of antibiotic-susceptible *K. pneumoniae* (Figure 4A). As with *S. aureus*, the effect of t-ZnO against *K. pneumoniae* did not appear to be concentration-dependent. The reduction in bacterial numbers was stronger than with the s-ZnO particles and was able to trigger complete eradication even at low concentrations. In contrast, the effect of the t-ZnO against the multidrug-resistant *K. pneumoniae* strain remains rather weak, and the s-ZnO have a much stronger effect. The methylene blue treatment of s-ZnO and t-ZnO did not have an enhancing effect, and the reduction was even weaker than with the ZnO formulations without methylene blue treatment.

### 2.3. Activity against Pseudomonas aeruginosa and Enterococcus faecalis

Treatment of *Enterococcus* (*E.*) *faecalis* and *Pseudomonas* (*P.*) *aeruginosa* cultures with t-ZnO did not exhibit significant reductions in bacterial counts at any concentration after 3 or 24 h compared with the untreated controls in orientating screening tests (Figure 2C,D). Thus, these attempts were not further pursued.

## 3. Discussion

Infectious diseases have not yet been overcome. The causative agents, which may be viruses, fungi, parasites, or bacteria, have been shown to be versatile in responding to measures used by humans to prevent or treat disease. In response to the evolutionary pressure during anti-infective therapy, antibiotic-resistant bacteria have either evolved or were already preformed and have been able to spread in the community. Thus, the arsenal of antimicrobials that are safely effective and tolerated is becoming increasingly narrow.

Recent analyses of health data showed that six pathogens were responsible for nearly 80 % of deaths due to antibiotic-resistant bacteria in 2019 [24,25]. From this group of clinically relevant microorganisms, we selected *S. aureus, K. pneumoniae, P. aeruginosa,* and additionally *E. faecalis* for testing with ZnO particles.

*S. aureus* has unusually high environmental stability and is insensitive to desiccation, temperatures up to 60 °C, acidic environments, and high sodium chloride concentrations. This favors its spread in the context of hospitalization. *S. aureus* is frequently found as a colonizing bacterium in the nasopharynx but also triggers severe infectious diseases, such as wound infections, abscesses, bone and joint infections, endocarditis and consequently, bloodstream infections [26]. In addition, there are toxin-mediated diseases caused by *S. aureus*, such as toxic shock syndrome, staphylococcal scalded skin syndrome, and food poisoning. *S. aureus* shows a broad spectrum of antibiotic resistance, from strains sensitive to all beta-lactams up to methicillin resistance, which renders almost all standard beta-lactam antibiotics ineffective. In almost all other antibiotic classes, resistance is also frequently observed [27].

*K. pneumoniae* is a species of Enterobacterales that is found ubiquitously, including in the human gut and, upon pneumonia, eventually also in the respiratory tract. Especially in hospitalized and immunocompromised individuals, *K. pneumoniae* frequently cause urinary tract infections and soft-tissue infections and may result in bloodstream infections and pneumonia cases [28]. *K. pneumoniae* is intrinsically resistant to penicillins, and increasingly, however, plasmid-encoded resistance to other beta-lactams or resistance to all cephalosporins is emerging. The increased occurrence of carbapenemases is threatening, which also renders the remaining carbapenem antibiotics ineffective [29]. Strains resistant to fluoroquinolones and aminoglycosides are also frequently detectable.

As a bacterium of the wet environment, *P. aeruginosa* can be found almost ubiquitously in the environment. Ventilator-associated pneumonia and urinary tract infections caused by *P. aeruginosa* occur during hospitalization [30]; wound infections caused by *P. aeruginosa* are frequent and dangerous complications for burn patients [31]. In addition, *P. aeruginosa* is a chronic pathogen in individuals with cystic fibrosis [32]. Already the wild type of *P. aeruginosa* has manifold resistances, and thus, only a limited number of antibiotics can be considered [33]. In addition, resistance to further antibiotics (piperacillin, ceftazidime, cefepime, carbapenems, aminoglycosides, and quinolones) can develop through mechanisms such as betalactamases, efflux pumps, mutation of porins, and carbapenemases.

*E. faecalis* resists temperatures up to 60 °C, high salt concentrations, and alkaline environments and belongs to the core of the gut microbiome. Hospitalized and immunocompromised individuals are particularly susceptible to *E. faecalis* infections, which present as urinary tract infections, endocarditis, and also wound infections [34,35]. It is common for bloodstream infections to be caused by such infectious foci [36,37]. The genus Enterococcus is intrinsically resistant to cephalosporins, while resistance to aminopenicillins and glycopeptides may occur, also in *E. faecalis* [37].

For the discovery of new antibiotics, various strategies, such as natural products [38,39] or the use of computational design [39,40], are applied. The combination of microbiology and materials science is aimed at developing new antimicrobial substances to be added to the therapeutic arsenal. The interest in nanosurface formulations of ZnO as active ingredients is very high due to the advantageous properties of these particles, and various possible applications have already been described [9]. The method of production by flame-transfer synthesis and the resulting structure set the t-ZnO apart from the rest of the s-ZnO particles [9,11,12,22,23].

In this study, a time-dependent killing of bacteria could be observed, with remarkable differences between the bacterial species and even resistance phenotypes. t-ZnO and s-ZnO particles showed advantages and disadvantages depending on species and strain.

While *S. aureus* showed a cell-number reduction of an average of three orders of magnitude at 1 mg/mL after 24 h of incubation, *K. pneumoniae* showed a similar cell-number reduction with 2 mg/mL. s-ZnO particles exhibited partly stronger reductions of bacterial counts in *S. aureus* compared to the t-ZnO. The methylene blue-treated ZnO particles were also shown to be effective against MRSA. The untreated particles and tetrapods and the methylene blue-treated tetrapods showed a comparable effect on methicillin-sensitive and methicillin-resistant *S. aureus*. However, the t-ZnO were more successful than the s-ZnO particles in eradicating *K. pneumoniae*. Compared to the s-ZnO particles, a lower concentration of t-ZnO was required. t-ZnO did not show effects on *P. aeruginosa* and *E. faecalis* strains. Thus, t-ZnO displayed considerable antibacterial activity against *S. aureus* and *K. pneumoniae* isolates.

Mechanisms of the antibacterial activity of metallic nanomaterials have been investigated successfully in various application systems [41,42,43,44]. Among these, one can find photothermal effects, *i.e.,* heating up of the nanoparticles by light exposure and causing a local temperature rise that can kill bacteria. A second light-triggered effect is the selective dissolution of the metal nanomaterial compounds and the resulting cell damage from one of the manifold metal-bacteria interactions. Thirdly, photochemically induced reactive oxygen species (ROS) from the nanoparticles have been described. Here, the semiconducting properties of metal oxide nanomaterials were used, where the generation of electron-hole pairs with an elevated energy level can lead to the oxidation and reduction of media or the particle itself and invoke ROS [2].

t-ZnO adds an additional principle of action to the spectrum of antibacterial mechanisms, which is independent of conventional soluble antibiotics and also at least partially different from that of small nanomaterials. The t-ZnO particles have a size of up to 30 µm, while the s-ZnO particles have a size in the nano- to micrometer range. Due to their size, t-ZnO will not be taken up by eukaryotic cells and will not penetrate into them. However, s-ZnO particles can be taken up by eukaryotic cells and induce oxidative stress by reactive oxygen species and cell death [19,21,45]. Compared to s-ZnO particles, the t-ZnO exhibit up to 270-fold higher effective concentration values for half-maximal effects (EC_50_) and, thus, significantly lower cytotoxicity [16]. The comparable effect of t- and s-ZnO particles and the lower cytotoxicity of t-ZnO compared to s-ZnO particles makes tetrapods an interesting drug candidate with a suitable therapeutic window. Surface modifications of the t-ZnO, such as activation by methylene blue treatment, might allow alterations of the host range of antibacterial activity.

The results lead to the speculation that the mode of action of t-ZnO, unlike s-ZnO, is not primarily concentration-dependent. This may indicate a different mechanism of action for t-ZnO than for s-ZnO. Due to the more distinct crystalline structure of t-ZnO, the contact effect against bacteria could be stronger. In contrast, the s-ZnO could release more reactive oxygen species or zinc ions due to their higher surface area. Although the exact antimicrobial mechanism of action of t-ZnO is still unknown, given the same chemical composition as s-ZnO particles, there are several possibilities: electrostatic interaction with the tetrapod arms, formation of reactive oxygen species, or the release of zinc ions [46,47]. Some limitations should also be mentioned, for example, the fact that ZnO is not soluble in watery liquids, resulting in suspensions that are difficult to handle because of the tendency of ZnO to sediment quickly, which makes it difficult to establish precise concentrations. Although the crystalline form of t-ZnO is unusual when handled in aqueous solutions, t-ZnO, with its surface chemistry, might provide a new interesting mode of action in combating bacterial infections in the future. Moreover, non-soluble compounds may be advantageous for local treatment, such as for wounds [9].

t-ZnO can be further optimized in several ways. During ultraviolet light treatment with methylene blue, ZnO tetrapods react photocatalytically with the dye, and the reactive surface of the tetrapods is expanded [47]. In addition, active ingredients can be attached to the t-ZnO surface in order to release them in a controlled manner [9]. This combination of antimicrobial activity and drug release could be advantageous in the care of chronic wounds or superficial infections. The low cytotoxic effect and the modification possibilities [45] are features of t-ZnO that are advantageous for antimicrobial agents.

**Conclusion:** Taken together, ZnO tetrapods showed considerable bactericidal activities against both *Staphylococcus aureus* and *Klebsiella pneumoniae* isolates, including multi-resistant strains. In *Staphylococcus aureus* at 0.5 mg/mL and *Klebsiella pneumoniae* at 0.25 mg/mL, an almost complete elimination was reached after 24 h. By surface modification, the activity of spherical ZnO particles against *Staphylococcus aureus* was even enhanced. In contrast to antibiotics, ZnO particles are largely insoluble and less concentration-dependent. Thus, we describe an extension to the spectrum of antibacterial principles against *Staphylococcus aureus* and *Klebsiella pneumoniae.*

## 4. Materials and Methods

**Bacteria:** *S. aureus* ATCC 6538, methicillin-resistant *S. aureus* ATCC 33593, *E. faecalis* (ATCC 29212), and *P. aeruginosa* (ATCC 27853), *K. pneumoniae* (ATCC 4352), and *K. pneumoniae* NTC 13438 expressing KPC3 carbapenemase (resistant against third-generation cephalosporins, fluoroquinolones and carbapenems) were purchased from the American type culture collection (ATCC, Manassas, VA, USA).

**Zinc Oxide Particles:** s-ZnO particles were purchased from Carl Roth GmbH, Karlsruhe, Germany. t-ZnO were produced via the flame transport synthesis as previously described [11,12,22,23]. In brief, zinc microparticles (particle size < 1 µm) supplied by Carl Roth were mixed with polyvinylbutyral (PVB) and heated inside a ceramic crucible to approximately 900 °C. During the following 30 min, the PVB burned, capturing the oxygen and protecting the zinc from oxidation until it started to evaporate. The reaction with oxygen then took place in the gas phase, and the ZnO tetrapods formed.

**Methylene blue treatment:** The t-ZnO and s-ZnO particles were irradiated with UV light (7.5 mW/cm^2^) for 3 h in water containing 10 µM methylene blue O (Sigma Aldrich, Merck, Darmstadt, Germany) [48], washed, and then used for efficacy testing.

**Statistical analysis** was performed using GraphPad Prism 9 and Microsoft Excel software. The figures were generated with Microsoft Excel 2013.

**Electron microscopy.** Scanning electron microscopy (SEM) was performed on a Zeiss Supra 55 VP (Zeiss, Ulm, Germany) [9,11,12,22]. The t-ZnO and s-ZnO were investigated with an acceleration voltage of 5 kV.

**Susceptibility testing:** Killing rates of different concentrations of tetrapods and amorphous s-ZnO particles against Gram-negative and Gram-positive bacteria were determined as described previously [49].

In brief, one fresh bacteria colony from a Columbia sheep blood agar plate (Thermo Scientific, Waltham, MA, USA) was inoculated into 10 mL tryptic soy broth (TSB) (Sigma Aldrich, Munich, Germany) and incubated for 12 h at 37 °C. Fifty μL of the culture was added to 10 mL of TSB/10 mM NaCl and incubated for 3 h at 37 °C. The bacteria suspension was diluted with 10 mM NaCl to reach an optical density (600 nm) of 10^5^ colony forming units (cfu/mL). Next, the microbial suspension was added to suspensions of either t-ZnO or s-ZnO particles in phosphate buffer with 1% TSB broth. Growth controls in buffer without t-ZnO or s-ZnO particles were performed in parallel for each experiment (Sigma Aldrich, Germany).

Depending on experimental settings, samples were taken at the indicated time points, and the number of cfu/mL were determined by the plate count method. The bacteria were diluted with 0.85% NaCl solution in serial 10-fold dilutions, and 100 μL of these dilutions were spread on lysogeny broth agar plates. The plates were incubated overnight at 37 °C, and the colonies were counted. The values of cfu/mL were calculated from the average of two separate agar plates.

## Figures and Tables

**Figure 1 ijms-24-03444-f001:**
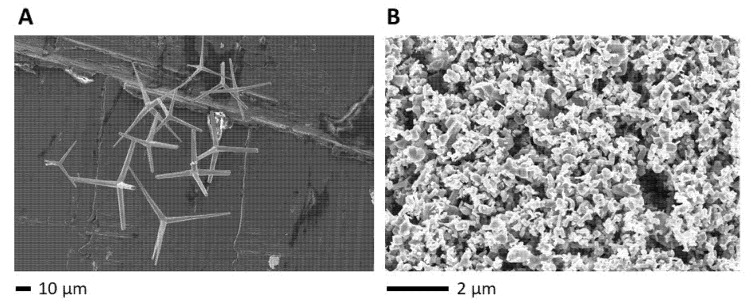
Electron micrographs of (**A**) ZnO tetrapods (t-ZnO) and (**B**) commercially available s-ZnO particles. s-ZnO particles are shown, which are supposed to be the major functional.

**Figure 2 ijms-24-03444-f002:**
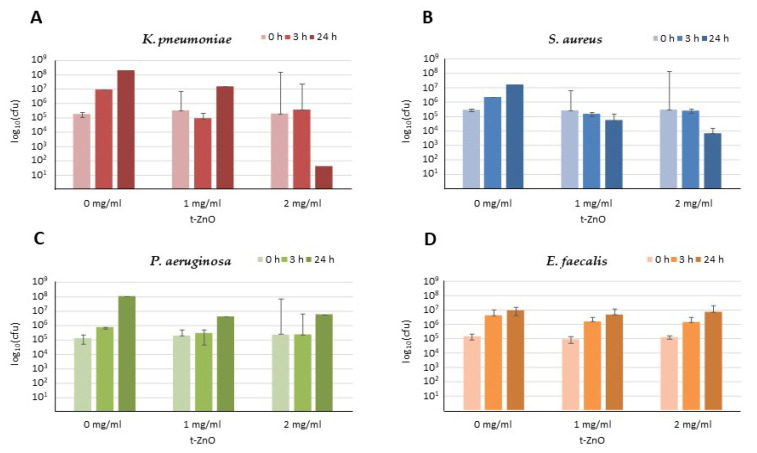
Orientating antibacterial testing of untreated ZnO tetrapods against (**A**) *K. pneumoniae*, (**B**) *S. aureus*, (**C**) *P. aeruginosa*, and (**D**) *E. faecalis* in means of independent triplicate values with standard deviations. Drug concentrations are shown in mg/mL. cfu = colony forming units.

**Figure 3 ijms-24-03444-f003:**
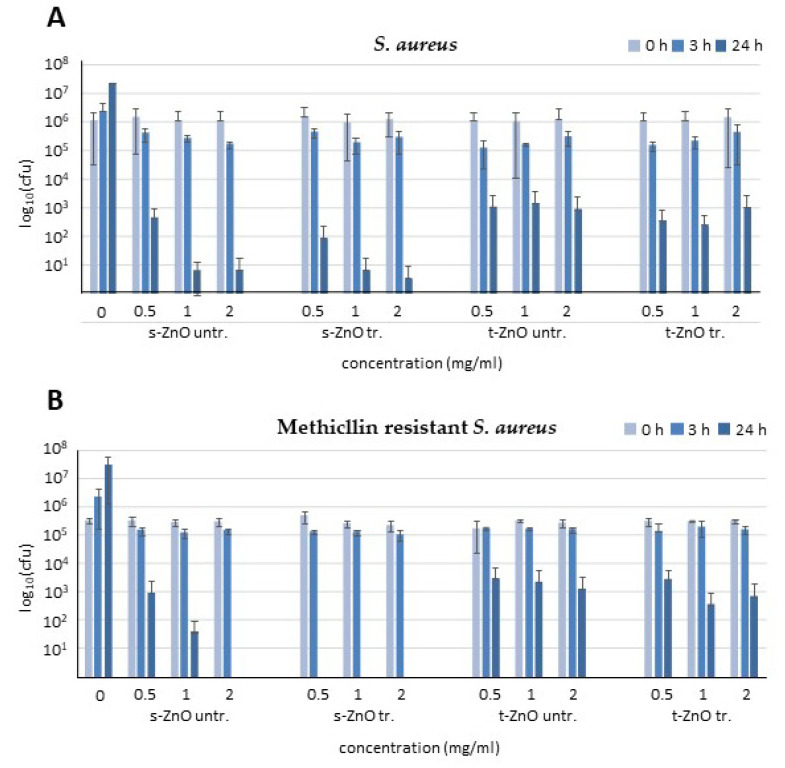
Antibacterial efficacy testing of untreated s-ZnO particles (s-ZnO untr.), methylene blue-treated particles (s-ZnO tr.), untreated ZnO tetrapods (t-ZnO untr.), and methylene blue-treated ZnO tetrapods (t-ZnO tr.) against (**A**) *Staphylococcus* (*S.*) *aureus* and (**B**) methicillin-resistant *S. aureus* (MRSA). Data are presented as mean values from three independent experiments (*n* = 3) with standard deviations. Drug concentrations are shown in mg/mL. cfu = colony forming units.

**Figure 4 ijms-24-03444-f004:**
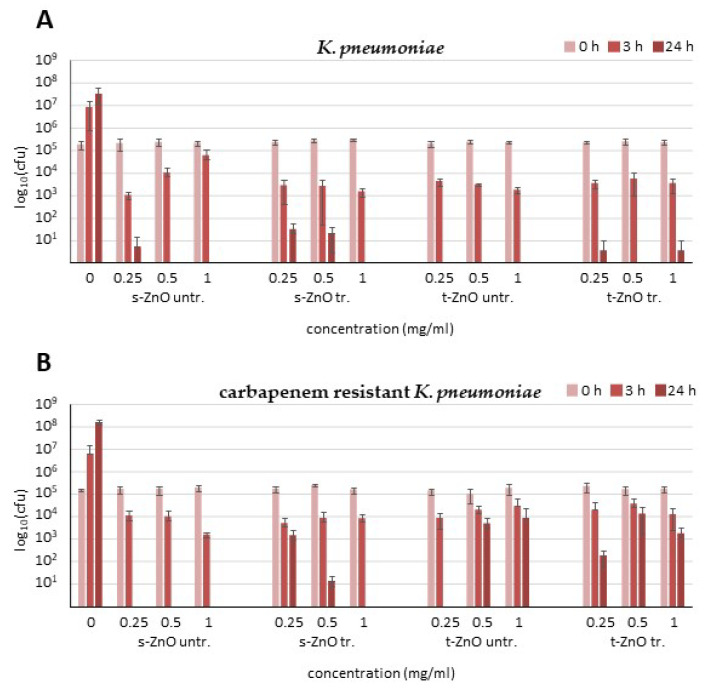
Antibacterial efficacy testing of untreated s-ZnO particles (s-ZnO untr.), methylene blue-treated particles (s-ZnO tr.), untreated ZnO tetrapods (t-ZnO untr.), and methylene blue-treated ZnO tetrapods (t-ZnO tr.) against (**A**) carbapenem-sensitive *K. pneumoniae* and (**B**) carbapenem-resistant *K. pneumoniae*. Data are presented as mean values from three independent experiments (*n* = 3) with standard deviations. Drug concentrations are shown in mg/mL. cfu = colony forming units.

## Data Availability

Not applicable.

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
