# Peer review of "Antibacterial Activity of Nanostructured Zinc Oxide Tetrapods"

_ijms, 2023, doi:10.3390/ijms24043444_

Round 1
Reviewer 1 Report
This is a very interesting work that connects the material and medical science. This paper investigated the antibacterial and bactericidal properties of ZnO tetrapods (t-ZnO) in comparison to spherical ZnO (s-ZnO) nanoparticles. The time-dependent killing of bacteria could be observed, with remarkable differences between the bacterial species and even resistance phenotypes. The t-ZnO and s-ZnO nanoparticles showed advantages and disadvantages depending on species and strain, which is not surprising. The main shortage of this paper is the lack of a mechanism proposed for ZnO action which is supposed to be different from antibiotics. However, I have just minor remarks:
1. Missing measurements in Figure 2, after 3h for C) P. aeruginosa
2. Please define EC50 values for non-medical readers
Author Response
Thank you very much for the rapid review process and for the highly valuable comments of the two reviewers. Accordingly we have performed additional experiments, especially for improving figure 2. Moreover, we have thoroughly worked on the manuscript and we hope that we have met the issues raised by the referees.
Reviewer 1
- Missing measurements in Figure 2, after 3h for C) P. aeruginosa
The experiments have been repeated independently and the figure has been revised accordingly.
- Please define EC50 values for non-medical readers
The toxicological term is now defined (effective concentration).

Reviewer 2 Report
In this manuscript, Büter et al. have described the Zinc oxide tetrapods preparation, characterization, and antibacterial properties against Staphylococcus aureus, Klebsiella pneumonia, Pseudomonas aeruginosa, and Enterococcus faecalis. But there are many limitations and gaps in the study conducted. I would suggest reconsidering after major revisions. I have listed major concerns and must revisions below.
1. Please state the novelty of this study with clarity.
2. ZnO terapods look like microparticles but stated as nanostructures please provide proof for the size of the particles (maybe DLS).
3. An extensive English edition is a must. Many typos and grammatical errors.
4. Affiliations are repeated please revise.
5. The abstract should be rewritten and major results and concentration should be added. Lines 29-30 should be elaborately explained (mechanism of action).
6. In the introduction, please add more references and please write in order what, why how and in your hypothesis with examples. Please refer and follow these articles to write an introduction https://pubs.acs.org/doi/full/10.1021/acssuschemeng.2c02457 https://www.ncbi.nlm.nih.gov/pmc/articles/PMC8405884/ https://www.sciencedirect.com/science/article/pii/S0304389420326078
7. Lines 44-47 elaborate as we are talking about antibacterial activity, not antiviral activity. please provide suitable examples to the context.
8. Rewrite and revise lines 36-63.
9. Please change Figure 1 and provide more SEM images. Rewrite the captions. Provide more characterization data and elaborate on the size ZnO tetrapods.
10. In Figure 2 the period is from 0h,3h,24h. There is not making a clear indication of the working of nanoparticles and antibacterial properties. The leap from 3-24h is a big gap where antibacterial effect is calculated. Please kindly provide the reason why the time such as 6h,12h results where not there. In figure 2A the effect of NPs on K. pneumoniae at 24h and 1mg/ml shows no death and 2mg/ml at 24h its killed is relatable. But can you explain why at 3h the killing effect is same in both concentration as it is highly confusing.
11. Figure 2C the group with 0mg/ml there is no growth at 3h is highly surprising as that is not possible. Please explain how this is possible when there is growth at 24h, this is highly confusing when compare with the other concentrations.
12. In the whole figure 2 there is no error bar, please add the error bars.
13. There are no complete characterization studies. Please include those (eg: TEM, FTIR, XRD etc.)
14. What is the hypothesis of using methylene blue coupled treatment with the NPs, please explain the mechanism and role of methylene blue in the treatment.
15. Discussion is insufficient and vague to prove the claims of the authors, please add some adequate references to support the findings. Please follow the references that might help
https://www.nature.com/articles/nrmicro3028 https://www.mdpi.com/2624-8549/2/2/26/pdf?version=1589013555 https://www.mdpi.com/2079-4991/10/4/643/html https://www.sciencedirect.com/science/article/pii/S2452199X21002024
16. Add a conclusion and summarize the key results and findings.
Author Response
thank you very much for the rapid review process and for the highly valuable comments of the two reviewers. Accordingly we have performed additional experiments, especially for improving figure 2. Moreover, we have thoroughly worked on the manuscript and we hope that we have met the issues raised by the referees.
Reviewer 2
- Please state the novelty of this study with clarity.
The respective text sections have been sharpened.
- ZnO tetrapods look like microparticles but stated as nanostructures please provide proof for the size of the particles (maybe DLS).
Additional references for the structural properties have been added and the wording has been adjusted. It is indeed better to describe particles (instead of nanoparticles), however, with nano-structured surfaces.
- An extensive English edition is a must. Many typos and grammatical errors.
We have improved the wording at numerous positions, mostly with minor changes.
- Affiliations are repeated please revise.
The affiliations were originally prepared according to the rules of the journal. However, we have now reduced the redundancy.
- The abstract should be rewritten and major results and concentration should be added. Lines 29-30 should be elaborately explained (mechanism of action).
The abstract has been modified accordingly.
- In the introduction, please add more references and please write in order what, why how and in your hypothesis with examples. Please refer and follow these articles to write an introduction https://pubs.acs.org/doi/full/10.1021/acssuschemeng.2c02457, https://www.ncbi.nlm.nih.gov/pmc/articles/PMC8405884/, https://www.sciencedirect.com/science/article/pii/S0304389420326078
Thank you very much for the helpful suggestions which have been included.
- Lines 44-47 elaborate as we are talking about antibacterial activity, not antiviral activity. please provide suitable examples to the context.
The text has been adjusted accordingly. Only in the context of antiviral activity, there are some directly relevant observations available. Concerning the antibacterial effects, the situation is less defined and has been described accordingly.
- Rewrite and revise lines 36-63.
The text has been adjusted accordingly.
- Please change Figure 1 and provide more SEM images. Rewrite the captions. Provide more characterization data and elaborate on the size ZnO tetrapods.
More citations (with more EM images) have been added which also include reports on the physical properties.
- In Figure 2 the period is from 0 h, 3 h, 24 h. There is not making a clear indication of the working of nanoparticles and antibacterial properties. The leap from 3-24h is a big gap where antibacterial effect is calculated. Please kindly provide the reason why the time such as 6h,12h results where not there. In figure 2A the effect of NPs on K. pneumoniae at 24h and 1mg/ml shows no death and 2mg/ml at 24h its killed is relatable. But can you explain why at 3h the killing effect is same in both concentration as it is highly confusing.
We chose the time points 3 and 24 hours in order to be able to comment on rather early and on delayed effects. It is obvious, that there are only minor immediate effects, and the major effects are seen at a later stage. The period 24 hours is frequently used in medical microbiology. Of course it will be interesting to further determine the kinetic behavior. However, we needed to restrict the possible variables since the work load increases considerably with each additional parameter. The rather concentration-independent situation for the tetrapods has been discussed.
- Figure 2C the group with 0mg/ml there is no growth at 3h is highly surprising as that is not possible. Please explain how this is possible when there is growth at 24h, this is highly confusing when compare with the other concentrations.
On the basis of repeated experiments, the figure has been revised accordingly.
- In the whole figure 2 there is no error bar, please add the error bars.
On the basis of repeated experiments, the figure has been revised accordingly.
- There are no complete characterization studies. Please include those (eg: TEM, FTIR, XRD etc.)
More citations have been added which also include several reports on the physical properties.
- What is the hypothesis of using methylene blue coupled treatment with the NPs, please explain the mechanism and role of methylene blue in the treatment.
The hypothesis concerning the methylene-blue treatment has been included and discussed in the text.
- Discussion is insufficient and vague to prove the claims of the authors, please add some adequate references to support the findings. Please follow the references that might help: https://www.nature.com/articles/nrmicro3028, https://www.mdpi.com/2624-8549/2/2/26/pdf?version=1589013555, https://www.mdpi.com/2079-4991/10/4/643/html https://www.sciencedirect.com/science/article/pii/S2452199X21002024
Thank you very much for the helpful suggestions which have been included.
- Add a conclusion and summarize the key results and findings.
The text has been adjusted accordingly.
Thank you very much for reconsidering our manuscript.

Round 2
Reviewer 2 Report
The authors have made noticeable changes overall but did not address the comments satisfactorily and in a detailed manner.
In the abstract, please explain the term solid body chemistry.
The graphical representation of all figures looks like processed using MS excel, not GraphPad prism 9. Please address this issue.
Please include the conclusion section separately.
Please briefly describe and include the ZnO tetrapod synthesis in Materials and methods.
On Line 308, "suspensions of t‐ZnO or s‐ZnO particles..." is it or or and?
Please include the graphical abstract for a better understanding of the audience.
More Characterization studies are required.
Lines 240-241 please elaborate and explain the mechanism of action with examples.
Author Response
Dear Editors,
thank you very much for the additional rapid review process with the highly valuable comments of one reviewer. In order to describe the spherical ZnO particles in better quality, we have performed additional electron microscopy studies. We have optimized the respective parts of the manuscript according to the comments and we hope that we have met the issues raised by the referee.
1. In the abstract, please explain the term solid body chemistry.
This has been commented accordingly.
2. The graphical representation of all figures looks like processed using MS excel, not GraphPad prism 9. Please address this issue.
Both programs have been used. This has been commented accordingly.
3. Please include the conclusion section separately.
A short separate conclusion section has been added.
4. Please briefly describe and include the ZnO tetrapod synthesis in Materials and methods.
This has been described accordingly.
5. On Line 308, "suspensions of t‐ZnO or s‐ZnO particles..." is it or or and?
This has been specified accordingly.
6. Please include the graphical abstract for a better understanding of the audience.
A graphical abstract has been added accordingly.
7. More Characterization studies are required.
The characterization studies have been described in more detail and another reference has been added.
8. Lines 240-241 please elaborate and explain the mechanism of action with examples.
This has been added accordingly.
Thank you very much for reconsidering our manuscript.
With kind regards, Helmut Fickenscher

Round 3
Reviewer 2 Report
Accept